# Fusing Deep Neural Networks with Multi-PDE Solvers: An Enhanced Approach to Battery Modeling

## Abstract

Accurately characterizing the migration behaviors of microscopic particles within lithium batteries is of great significance for the design and development of new battery technologies. During charge and discharge cycles, the intercalation and deintercalation of lithium ions between the cathode and anode involve complex multi-physics coupling processes. Traditional numerical methods often suffer from low computational efficiency when simulating such dynamic systems. To address this challenge, we propose a novel deep learning framework that directly incorporates the dynamic evolution of the system into the neural network architecture. Our approach ensures reliable parameter estimation and voltage reconstruction while significantly improving computational inference efficiency. Extensive simulation experiments validate the effectiveness of the proposed method, offering a new technical pathway for high-precision and efficient battery modeling.

## 1 Introduction

Recent advancements in battery technology have underscored the necessity for accurate modeling of electrochemical processes, as these systems are pivotal for applications ranging from electric vehicles to renewable energy storage (Miao et al., 2019; Raja et al., 2021; Yin & Zhang, 2023; Hussain et al., 2025; Yeregui et al., 2025). Among the various approaches to battery modeling, the Pseudo-Two-Dimensional (P2D) model has emerged as a prominent framework for simulating the complex dynamics of lithium-ion batteries. The P2D model effectively captures the intricacies of mass transport, charge transfer, and thermal behavior within the battery, governed by a set of partial differential equations (PDEs) (Doyle & Newman, 1995; Jokar et al., 2016). However, solving these PDEs poses significant challenges, particularly in estimating key parameters that influence battery performance, such as diffusion coefficients, reaction rates, and conductivity (Jokar et al., 2016; Han et al., 2021).

Traditional numerical methods for solving PDEs in the context of battery modeling often rely on discretization techniques, such as finite difference (Han et al., 2021; Chen et al., 2023) or finite element methods (Han et al., 2015; Pang et al., 2021). While these methods can provide accurate solutions, they are computationally intensive and may struggle to adequately capture the underlying physics of the system, especially in high-dimensional parameter spaces. Moreover, these approaches often require extensive mesh generation and can suffer from convergence issues, leading to longer simulation times and increased resource consumption.

Therefore, some neural network-based methods have appeared to solve equations that contain differential structures or derivation structures (Ruthotto & Haber, 2020; Liu et al., 2024; Ye et al., 2024), while they cannot use the prior physical information such as the autograd mechanism. To address these limitations, we propose an innovative approach that takes advantage of physics-informed neural networks (PINN) (Raissi et al., 2017a;b). This emerging methodology integrates the governing equations directly into the neural network architecture, allowing for the simultaneous estimation of parameters while ensuring adherence to the physical laws governing the system. By embedding the physics into the training process, PINN can learn the underlying dynamics of the battery system without the need for extensive computational resources typically associated with traditional numerical methods.

In this work, we present a comprehensive framework that enhances battery modeling through the application of deep neural networks to assist in solving the multi-PDEs inherent in the P2D model. Our approach improves accuracy while significantly reducing the computational burden associated with conventional methods. By training the neural network to minimize the discrepancy between predicted and observed data while adhering to physical constraints, we enable a more efficient exploration of the parameter space. This method allows us to effectively estimate critical parameters that govern battery behavior, leading to enhanced predictive capabilities and insights into battery performance under various operating conditions.

The main contributions of our work are given below: (1) We develop a novel framework that integrates PINN into the P2D battery modeling, providing a more efficient method for solving complex PDEs. (2) Our approach facilitates the simultaneous estimation of critical parameters governing battery dynamics, improving the accuracy of predictions while reducing computational costs. (3) The framework is designed to be adaptable to various battery chemistries and configurations, making it a powerful tool for researchers and engineers working in the field of battery technology.

## 2 RELATED WORKS

**Previous Battery Models** such as the Equivalent-Circuit Model (ECM) (Yann Liaw et al., 2004) and the Single Particle Model (SPM) (Guo et al., 2010) have significantly contributed to the understanding of lithium-ion battery dynamics. The ECM simplifies the battery's electrochemical processes into an electrical circuit, allowing for efficient simulations and real-time parameter estimation. However, it may overlook the complex interactions between solid and electrolyte phases (Guo & Shen, 2021; Hua et al., 2021). In contrast, the SPM represents active material in electrodes as a single spherical particle, accounting for lithium ion diffusion and reaction kinetics (Namor et al., 2017). While it provides more detailed insights into particle behavior, it simplifies the system by neglecting interactions among multiple particles. These foundational models have paved the way for more advanced approaches, such as the P2D model, which integrates macroscopic and microscopic perspectives for a comprehensive analysis of battery performance.

**PINN** Physics-Informed Neural Networks (PINN) stand out to unify the adaptability of data-driven models with the interpretability due to their abality to incorporate physical laws directly into the processing system (Raissi et al., 2017a;b). In the calculation process of loss function, they can embed the knowledge of any physical laws that govern a given data-set in the learning process, and can be described by partial differential equations (PDEs). Since most of the physical laws that govern the dynamics of a system can be described by partial differential equations, solving the governing partial differential equations of physical phenomena using deep learning has emerged as a new field of scientific machine learning. Also PINN allow for addressing a wide range of problems in computational science and represent a pioneering technology leading to the development of new classes of numerical solvers for PDEs. PINN can be thought of as a meshfree alternative to traditional approaches (e.g., CFD for fluid dynamics), and new data-driven approaches for model inversion and system identification. Notably, the trained PINN network can be used for predicting the values on simulation grids of different resolutions without the need to be retrained. In addition, being neural fields, they allow for exploiting automatic differentiation (AD) to compute the required derivatives in the partial differential equations, a new class of differentiation techniques widely used to derive neural networks assessed to be superior to numerical or symbolic differentiation.

## 3 METHODOLOGY

### 3.1 PSEUDE-TWO-DIMENSIONS

The P2D model developed by Doyle et al. (1993); Doyle & Newman (1995) effectively captures the dynamics of both the solid and electrolyte phases in lithium-ion batteries, specifically within the positive electrode, separator, and negative electrode. At the macroscopic level, it is assumed that the kinetics of the chemical reactions predominantly influence battery dynamics along the $x$-dimension (which means the direction along which lithium ions transfer through the electrolyte, connecting the different phases of the battery and facilitating the electrochemical reactions necessary for charge and discharge). Conversely, at the microscopic level, the model considers the solid particles in both

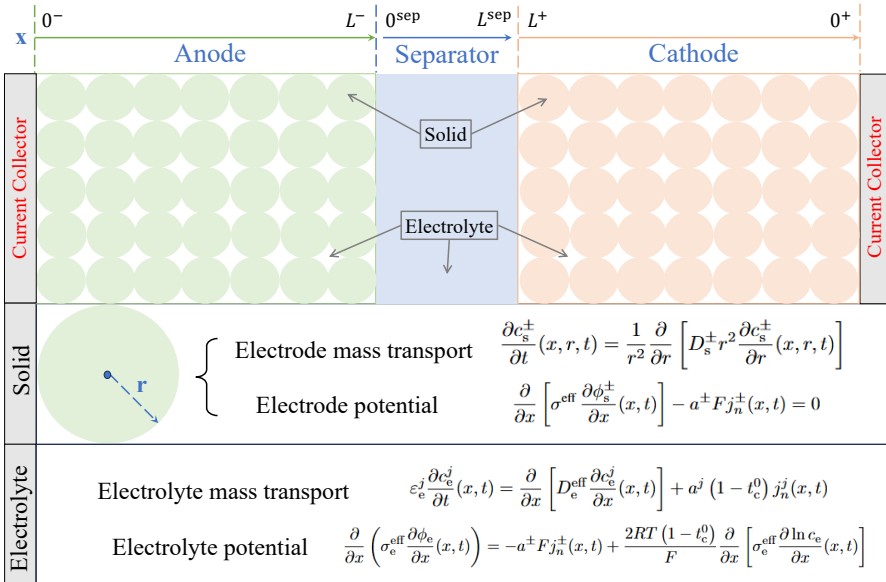

Figure 1: Schematic of electrochemical modeling for lithium-ion batteries. The upper axis represents the x-direction, along which lithium ions migrate through the electrolyte. The green and orange spheres denote the solid phase, while the remaining regions—including the separator—constitute the electrolyte phase. The figure also presents the partial differential equations governing mass transport and potential distributions in both the solid (electrode) and electrolyte phases.

electrodes as spherical entities with defined radii, enabling lithium ions to diffuse radially within the solid phase. This dual-scale approach allows for a comprehensive understanding of the intricate interplay between ionic and electronic transport processes in lithium-ion batteries.

As shown in the schematic of the P2D model in Fig. 1, the whole lithium-ion battery is divided into three different domains, i.e., anode (from $0^-$ to $L^-$), separator (from $0^s$ to $L^s$) and cathode (from $0^+$ to $L^+$). In P2D model, we suppose the condition that environment temperature is constant. In addition, the conductivity of the positive and negative electrode current collectors is extremely high, leading to no significant variation in the spatial coordinates along the $y$ (the thickness) and $z$ (the width of the battery cell) axes. In other words, the electrochemical reaction kinetics in the $x$-axis dimension dominate the battery's dynamic behavior. Additionally, to explain the intercalation and deintercalation behavior of lithium ions in the solid and liquid phases, it is assumed that there exists a single-dimensional solid spherical particle overlapping with the electrolyte, and spherical single particles are present in various regions between the positive and negative electrodes of the battery. The diffusion of lithium ions within the single-particle granules occurs in the radial $r$ dimension of the spherical particle, and information from the time dimension must also be considered. Therefore, this model is referred to as the multi-scale quasi-two-dimensional electrochemical P2D model. Considering the electrochemical kinetic behaviors in the solid and liquid phases as well as the electrochemical reactions at the phase interfaces, the kinetic equations for the lithium-ion battery P2D model are established based on porous electrode theory and concentration theory.

### 3.2 P2D EQUATIONS

In the Pseudo-2D (P2D) model—widely used for lithium-ion batteries—Mass Transport Equations, Potential Equations, and Butler-Volmer Equations work synergistically to characterize core electrochemical processes: **Mass Transport Equations** quantify the spatiotemporal distribution of lithium ions via solid-phase diffusion in electrodes, liquid-phase transport in electrolytes, and separator permeability, ensuring reactant availability and accounting for concentration polarization; **Potential Equations** describe charge flow by calculating solid-phase (electron conduction) and liquid-phase (ion conduction) potential gradients, establishing the driving force for transport and quantifying ohmic losses; **Butler-Volmer Equations** govern interfacial electrochemical reaction kinetics, link-

ing reaction rate to overpotential, capturing activation polarization, and bridging mass transport and potential distributions, together enabling comprehensive prediction of battery performance. For the boundary condition of PDE equations, we show these in Appendix A.2.

### 3.2.1 ELECTRODE MASS TRANSPORT

According to Fick (1855), the electrode mass transport function is shown as following:

$$\frac{\partial c_{\mathrm{s}}^{\pm}}{\partial t}(x, r, t) = \frac{1}{r^2} \frac{\partial}{\partial r} \left[ D_{\mathrm{s}}^{\pm} r^2 \frac{\partial c_{\mathrm{s}}^{\pm}}{\partial r}(x, r, t) \right]. \tag{1}$$

### 3.2.2 ELECTROLYTE MASS TRANSPORT

The the concentration of lithium ions:

$$\varepsilon_{\mathrm{e}}^{j} \frac{\partial c_{\mathrm{e}}^{j}}{\partial t}(x, t) = \frac{\partial}{\partial x} \left[ D_{\mathrm{e}}^{\mathrm{eff}} \frac{\partial c_{\mathrm{e}}^{j}}{\partial x}(x, t) \right] + a^{j} \left( 1 - t_{\mathrm{c}}^{0} \right) j_{n}^{j}(x, t), \tag{2}$$

where $D_{\mathrm{e}}^{\mathrm{eff}}$ is the effective electrolyte diffusion coefficient and it follows $D_{\mathrm{e}}^{\mathrm{eff}} = \varepsilon_{\mathrm{e}}^{1.5} D_{\mathrm{e}}$. $a = 3\varepsilon_{\mathrm{s}}/\Lambda_{\mathrm{s}}$ is the specific interfacial area.

### 3.2.3 SOLID POTENTIAL OF PARTICLES

The transform of solid potential $\phi_{\mathrm{s}}(x, t)$ is shown as:

$$\frac{\partial}{\partial x} \left[ \sigma^{\mathrm{eff}} \frac{\partial \phi_{\mathrm{s}}^{\pm}}{\partial x}(x, t) \right] - a^{\pm} F j_{n}^{\pm}(x, t) = 0, \tag{3}$$

where $\sigma^{\mathrm{eff}} = \varepsilon_{\mathrm{e}}^{1.5} \sigma^{\mathrm{e}}$ is the effective solid electrode conductivity.

### 3.2.4 ELECTROLYTE POTENTIAL

For the transform of electrolyte potential $\phi_{\mathrm{e}}(x, t)$:

$$\frac{\partial}{\partial x} \left( \sigma_{\mathrm{e}}^{\mathrm{eff}} \frac{\partial \phi_{\mathrm{e}}}{\partial x}(x, t) \right) = -a^{\pm} F j_{n}^{\pm}(x, t) + \frac{2RT \left( 1 - t_{\mathrm{c}}^{0} \right)}{F} \frac{\partial}{\partial x} \left[ \sigma_{\mathrm{e}}^{\mathrm{eff}} \frac{\partial \ln c_{\mathrm{e}}}{\partial x}(x, t) \right], \tag{4}$$

where $\sigma_{\mathrm{e}}^{\mathrm{eff}} = \varepsilon_{\mathrm{e}}^{1.5} \sigma_{\mathrm{e}}$ is the effective electrolyte conductivity and

$$\sigma_{\mathrm{e}} = 0.1297 \times (c_{\mathrm{e}}(x, t)/1000)^3 - 2.51 \times (c_{\mathrm{e}}(x, t)/1000)^{1.5} + 3.329 \times (c_{\mathrm{e}}(x, t)/1000).$$

### 3.2.5 BUTLER-VOLMER EQUATIONS

For interfacial electrochemical reaction kinetics, it follows:

$$\eta^{\pm}(x, t) = \phi_{\mathrm{s}}^{\pm}(x, t) - \phi_{\mathrm{e}}^{\pm}(x, t) - U^{\pm} \left[ \frac{c_{\mathrm{ss}}^{\pm}(x, t)}{c_{\mathrm{s,max}}^{\pm}} \right] - F R_{f}^{\pm} j_{n}^{\pm}(x, t),$$

$$j_{n}^{\pm}(x, t) = i_{0}^{\pm}(x, t) \left[ \exp \left( \frac{\alpha F}{RT} \eta^{\pm}(x, t) \right) - \exp \left( -\frac{\alpha F}{RT} \eta^{\pm}(x, t) \right) \right], \tag{5}$$

$$i_{0}^{\pm}(x, t) = k^{\pm} \left[ c_{\mathrm{e}}(x, t) \left( c_{\mathrm{s,max}}^{\pm} - c_{\mathrm{ss}}^{\pm}(x, t) \right) \right]^{\alpha_{\mathrm{a}}} \times \left[ c_{\mathrm{ss}}^{\pm}(x, t) \right]^{\alpha_{\mathrm{c}}},$$

$$c_{\mathrm{ss}}^{\pm}(x, t) = c_{\mathrm{s}}^{\pm} \left( x, R_{\mathrm{s}}^{\pm}, t \right),$$

where $R_{f}^{\pm}$ is the Solid Electrolyte Interface (SEI) film resistance, $U^{\pm}(\cdot)$ is the open-circuit potential (OCP) and $\alpha_{\mathrm{a}} = \alpha_{\mathrm{c}} = 0.5$.

### 3.3 PHYSICS-INFORMED NEURAL NETWORKS (PINN)

For our concentration, we adopt PINN strategy solving the combination of PDEs and Butler-Volmer equations. Compared with traditional methods, such as finite difference method (FDM) (Godunov & Bohachevsky, 1959), finite element method (FEM) (Bathe, 2007), and finite volume method (FVM)

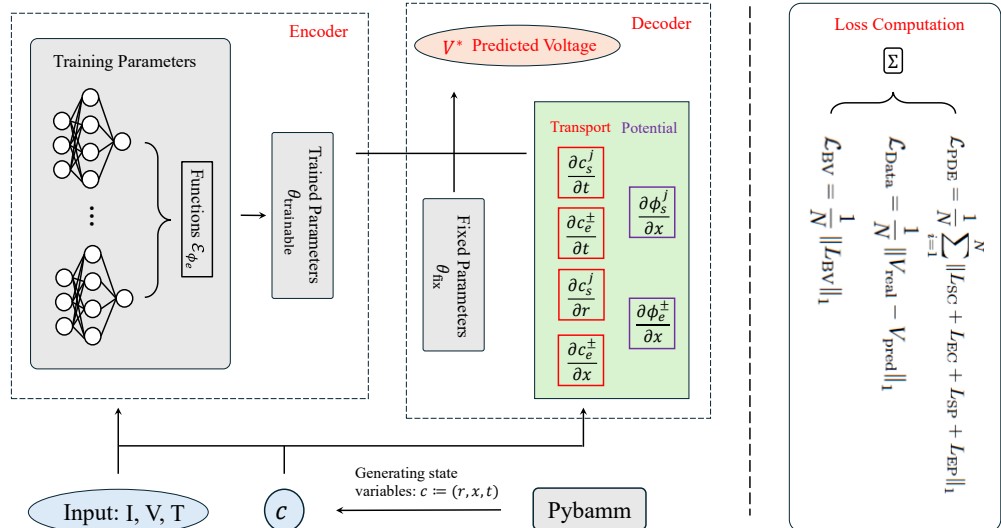

Figure 2: Overview of our model, comprising two main components: the encoder and the decoder. The encoder (left) generates learnable parameters with MLP based on the input variables $I, V, T$, representing current, voltage, and temperature, respectively. The decoder (right) utilizes the generated data from PyBaMM, which includes spatial coordinates $x, r$ for various time points, along with fixed and learnable parameters. It employs a PINN model to reconstruct the voltage and estimate the learnable parameters. And the loss function contains PDE loss, BL loss and Reconstruction loss.

(Eymard et al., 2000), which often face significant limitations, PINN model appeared as a solution about how to handle multi-scale and multi-physics problems.

To implement the PINN strategy for solving the combination of PDEs and Butler-Volmer equations, several key steps are typically involved. First, the mathematical model of the problem is formulated, including the governing PDEs, the Butler-Volmer equation for the electrochemical reaction, and the corresponding boundary and initial conditions. Second, a neural network architecture is selected. Common architectures include linear projection neural networks, deep residual networks (ResNets) (He et al., 2016), and recurrent neural networks (RNNs) (Sherstinsky, 2020), depending on the nature of the problem (e.g., static or dynamic, one-dimensional or multi-dimensional) (Raissi et al., 2017a;b; 2019). Third, the loss function of the PINN is constructed. The loss function usually consists of three parts: the residual loss from the PDEs (to ensure that the predicted solution satisfies the governing equations), the residual loss from the Butler-Volmer equation (to ensure that the electrochemical reaction kinetics are satisfied at the interface), and the reconstruction loss. Fourth, the PINN model is trained using an optimization algorithm (e.g., Adam (Kingma & Ba, 2014), LBFGS (Zhu et al., 1997)) to minimize the constructed loss function. During the training process, adaptive sampling techniques such as adaptive refinement of collocation points can be used to improve the training efficiency and the accuracy of the solution in regions with high gradients.

## 4 MODEL STRUCTURE

For our proposed solution, we adopt the PINN modeling strategy which leverages the strengths of both PINN and parameter estimation techniques. To begin with, we utilize an encoder structure which contains the linear projection neural network specifically designed for parameter estimation. Subsequently, we employ the PINN framework to refine our parameter estimates further. The PINN are particularly well-suited for this task because they incorporate physical laws and constraints directly into the learning process. By integrating the linear projection method with PINN, we create a robust and versatile modeling strategy. For our purpose, we aim to find the estimation of high sensitive parameters and reconstruct the voltage. The overview figure is shown in Fig. 2.

Table 1: Summary of parameters in P2D model, where the superscript $j \in \{+, -, \mathrm{sep}\}$.

| Category | Parameter | Unit | Description |
|---|---|---|---|
| Geometric parameters | $L^j$ | $m$ | Thickness |
| | $R^{\pm}$ | $m$ | Solid particle radius |
| | $A$ | $m^2$ | Electrode surface area |
| | $\varepsilon_s^{\pm}$ | $1$ | Solid active material volume fraction |
| | $\varepsilon_e^j$ | $1$ | Electrolyte volume fraction |
| Transport parameters | $D_s^{\pm}$ | $m^2 \cdot s^{-1}$ | Solid diffusion coefficient |
| | $D_e$ | $m^2 \cdot s^{-1}$ | Electrolyte diffusion coefficient |
| | $\sigma_s^{\pm}$ | $S \cdot m^{-1}$ | Electrode conductivity |
| | $t_c^0$ | $1$ | Transference number of lithium cation |
| Kinetic parameters | $\kappa^{\pm}$ | $m \cdot s^{-1}$ | Reaction rate coefficient |
| | $R_f^{\pm}$ | $\Omega \cdot m^2$ | SEI film resistance |
| Concentration parameters | $c_{s,\max}^{\pm}$ | $mol \cdot m^{-3}$ | Maximum ionic concentration |
| | $c_e^0$ | $mol \cdot m^{-3}$ | Initial electrolyte concentration |

## 4.1 ENCODER FOR PARAMETER ESTIMATION

In the methodology we propose, our framework concentrates on constructing a parametric mapping function—explicitly defined as the *encoder function*—to model the relationship between the system's measurable input variables (current, voltage, temperature) and the target latent parameter set $\theta_{\mathrm{trainable}}$. This encoder function, which is implemented based on a Multi-Layer Perceptron (MLP) architecture, is formally denoted as:

$$\theta = \mathcal{E}_{\phi_e}(I, V, T), \tag{6}$$

where: $\mathcal{E}$ represents the MLP-based encoder; $\phi_e$ denotes the trainable parameters of the MLP, including weights and biases across all hidden layers; $I$ is the input current; $V$ is the terminal voltage of the system; $T$ is the operating temperature; $\theta_{\mathrm{trainable}} = [\theta_{\mathrm{trainable}}^1, \theta_{\mathrm{trainable}}^2, \cdots, \theta_{\mathrm{trainable}}^n]^T$ is the vector of target latent parameters (e.g., solid diffusion coefficient) to be estimated.

The core objective of designing this MLP-based encoder $\mathcal{E}_{\phi}$ is to replace traditional linear mapping or heuristic parameter estimation methods with a data-driven, nonlinear framework. The MLP enables the encoder to learn and compress high-dimensional, context-dependent patterns between $I, V, T$ and $\theta_{\mathrm{trainable}}$. For example, in battery applications, the relationship between current pulses ($I$) and solid-phase lithium diffusion coefficients ($\theta_{\mathrm{trainable}}^1 = D_s$) is inherently nonlinear—high current densities induce non-uniform concentration gradients that linear models cannot represent, but the MLP's hierarchical structure can adaptively fit such dynamics. Also as a question of parameter estimation, it is enough to integrate activation layers into the MLP architecture of $\mathcal{E}$. Unlike shallow linear models, these activation functions introduce non-linearity at each hidden layer, enabling the encoder to learn complex mappings—such as the temperature-dependent saturation of reaction rates or the current-induced nonlinearity in electrolyte concentration gradients.

The complete set of parameters related to $\theta$ and the broader model are systematically categorized and detailed in Table 1. Notably, this table organizes parameters into four distinct groups to clarify their roles and properties which contains the trainable parameters directly generated by the encoder $\mathcal{E}_{\phi}(I, V, T)$ (i.e., the high-sensitivity $\theta_{\mathrm{trainable}}$ values learned from input data) and fixed, low-sensitivity parameters ($\theta_{\mathrm{fix}}$ e.g., material density, geometric constants) that are pre-determined via experimental calibration or literature and require no further optimization. This table also contains all parameters in the table accompanied by their respective units to ensure physical consistency. Importantly, both the encoder-derived trainable parameters $\theta_{\mathrm{trainable}}$ and the pre-fixed low-sensitivity parameters $\theta_{\mathrm{fix}}$ are collectively fed into the subsequent decoder layer.

## 4.2 DECODER FOR PINN TRAINING

This part of decoder structure is used to train the PINN model which is designed to output the dependent variable of interest, such as the cathode lithium ionic concentration $c_s$ at a specified time $t$,

solid particle point $r^{\pm}$ and anode or cathode or separator point $x^{\pm}, x^{\text{sep}}$. The model evaluates its predictive accuracy by comparing its outputs to the residuals derived from the underlying physical equations. In our case study, the PINN leverages the governing equations from the Pseudo-Two-Dimensional (P2D) model, allowing it to be trained to replicate the P2D model's behavior effectively. By utilizing a defined set of physical parameters—extracted from beginning-of-life experimentation or existing literature—the model is expected to accurately describe the P2D dynamics.

For the neural network component of the system, we have opted to combine a linear projection architecture with dense layers leading up to the output. Our methodology involves splitting the network into a branch network that processes input functions, such as lithium ionic concentrations $c$ and potential $\phi$, and a trunk network that handles collocation points, including time $t$, radius point $r$ and electrode point $x$. Therefore, the decoder structure can be simplified as following:

$$V_{\text{rec}} = \mathcal{D}_{\phi_d}(\theta_{\text{trainable}}, \theta_{\text{fix}}, x, r, t, I, V, T), \tag{7}$$

where $\theta_{\text{trainable}}$ and $\theta_{\text{fix}}$ are parameters given in encoder, $x, r$ are given by PyBaMM. To streamline the learning process, our PINN approach is structured to decouple the positive and negative electrodes. This separation simplifies the learning problem by allowing each electrode's neural network to focus on the losses associated with its respective domain's P2D equations. By treating each electrode independently, the PINN can effectively learn the unique dynamics of each domain. This configuration facilitates the study of specific mechanics, such as diffusion characteristics and aging mechanisms, in isolation. Additionally, this separation provides flexibility in refining the model for each electrode, thereby preventing the propagation of high-error cases across both domains.

The next crucial step involves defining the loss function necessary for training the PINN. In this initial phase, our objective is for the model to accurately replicate the behavior of the underlying P2D model. By effectively capturing the intricacies of the P2D dynamics, we aim to ensure that the PINN not only learns the fundamental relationships but also generalizes well to various operating conditions, paving the way for robust predictions in practical applications. Based on the P2D principles described in equations 1, 2, 3, 4 and 5, the total loss will be the sum of the terms that describe the battery model. Thus, the terms corresponding to the PDE Loss $\mathcal{L}_{\text{PDE}}$, Data Loss $\mathcal{L}_{\text{Data}}$ and Butler-Volmer Loss $\mathcal{L}_{\text{BV}}$:

$$\mathcal{L} = \mathcal{L}_{\text{PDE}} + \mathcal{L}_{\text{BV}} + \mathcal{L}_{\text{Data}}, \tag{8}$$

where

$$\mathcal{L}_{\text{PDE}} = \frac{1}{N} \sum_{i=1}^{N} \|L_{\text{SC}} + L_{\text{EC}} + L_{\text{SP}} + L_{\text{EP}}\|_1,$$

$$\mathcal{L}_{\text{BV}} = \frac{1}{N} \|L_{\text{BV}}\|_1,$$

$$\mathcal{L}_{\text{Data}} = \frac{1}{N} \|V_{\text{real}} - V_{\text{pred}}\|_1,$$

and $L_{\text{SC}}, L_{\text{EC}}, L_{\text{SP}}, L_{\text{EP}} and L_{\text{BV}}$ are absolute errors given by the PDE equations 1, 2, 3, 4 and 5 respectively. And $V_{\text{real}}$ is the real voltage given by data, while $V_{\text{pred}}$ is the reconstruction prediction of voltage given by trainable parameters.

## 5 EXPERIMENTS

### 5.1 DATA GENERATING

In this section, we first employed the Python Battery Mathematical Modeling (PyBaMM) framework (Sulzer et al., 2021) to construct the necessary data for our study. Given that our initial dataset comprises only current, voltage, and temperature, we recognized the need for more granular data to effectively utilize Physics-Informed Neural Networks (PINN). To address this requirement, we fully simulated the reference battery within the PyBaMM environment. With the capabilities of the PyBaMM model, we are able to generate additional detailed data by simply providing key parameter information about the battery, as well as the specific charging and discharging conditions. Alongside the variation of current over time, this allows us to construct a comprehensive dataset that incorporates the solid particle point $r$ and anode or cathode point $x$ within the structures of the positive and negative electrodes, as well as the electrolyte. This approach not only enhances the richness of our training dataset but also significantly alleviates the burden on the amount of training data required.

To determine the overall battery voltage, we can compute the voltage difference between the positive and negative electrodes. This calculation enables us to reconstruct the battery voltage, effectively modeling the entire system. During the training process for the P2D model, we can represent the battery voltage at the moment when $x = 0$ by supplying all corresponding values, ensuring that our model captures the essential dynamics. Regarding the parameters used in our simulations, we adopted a dual approach. For certain fixed parameters, we maintained consistency by using the same values for both the PINN and the PyBaMM model. However, for the highly sensitive parameters that we intend to estimate, we utilized accurate values during the data generation process within PyBaMM. This strategic selection of parameters not only reinforces the validity of our model but also enhances its capability to learn and generalize effectively.

## 5.2 Experiment Settings

In our approach, we categorize the parameters of neural networks into two distinct parts: the linear projection for high-sensitivity parameter estimation and the parameters for training Physics-Informed Neural Networks (PINN). For the high-sensitivity parameter estimation, we utilize the *Tanh* activation function between two linear projection layers. In contrast, for the PINN training parameters, we adopt a multi-head linear projection as the deep neural network architecture, incorporating activation layers between the linear layers. All experiments were conducted using the PyTorch framework, leveraging its autograd functionality for automatic differentiation. Our models were trained on a local machine equipped with a V100 GPU and 32 GB of RAM.

## 5.3 Results

For parameter estimation and voltage reconstruction, we developed a Linear Projection architecture, as outlined in the methodology section. All hyperparameters and implementation details have been meticulously documented to ensure reproducibility. Our model is trained at a 1C rate, utilizing continuous current inputs, with random selections made after each epoch to enhance the training process. We train PINN-based model to undergo an extensive training regimen for 20,000 epochs. Leveraging a V100 GPU, we achieved an average training time of approximately 50 milliseconds per epoch. The reconstructed voltage response generated by the PINN model, when compared to the simulation data, is depicted in Fig. 3 (a), highlighting the differences in voltage. Furthermore, we present the voltage reconstruction at time points 0, 100 and 300 seconds, along with a boxplot illustrating the voltage error across 50 trials in Fig. 3 (b). Regarding the voltage reconstruction results, our method demonstrates the capability to achieve convergence within 10 mV, indicating substantial advancements in battery voltage estimation. Additionally, we also evaluated the parameter estimation performance of our model, and the detailed results will be presented in the following Table 2. The parameters estimated by our proposed method fall within acceptable tolerance ranges. For instance, if we treat the target value as the estimated value, we can achieve similar voltage reconstruction, with MSE of $1.29 \times 10^{-5}$. Furthermore, when compared to several other optimization methods, such as L-BFGS-B (99.77%) in Yeregui et al. (2025), we achieve a significant reduction in error (31.00%).

Table 2: Estimation of some parameters.

| Parameter | Target Value | Estimation Value | % |
|:---:|:---:|:---:|:---:|
| $D_{\mathrm{s}}^{+}$ | $1.00 \times 10^{-13}$ | $1.31 \times 10^{-13}$ | 31.00 |
| $D_{\mathrm{s}}^{-}$ | $3.90 \times 10^{-14}$ | $3.73 \times 10^{-13}$ | 4.36 |
| $D_{\mathrm{s}}^{\mathrm{sep}}$ | $7.50 \times 10^{-11}$ | $7.95 \times 10^{-11}$ | 6.01 |
| $\sigma_{\mathrm{s}}^{+}$ | 10 | 11.91 | 18.98 |
| $\sigma_{\mathrm{s}}^{-}$ | 215 | 201.09 | 6.47 |

## 6 Ablation Study

In our study, we primarily focused on the network layer structure of PINN models and the loss function related to PDEs. Additionally, we compared our method with a simpler autoencoder structure

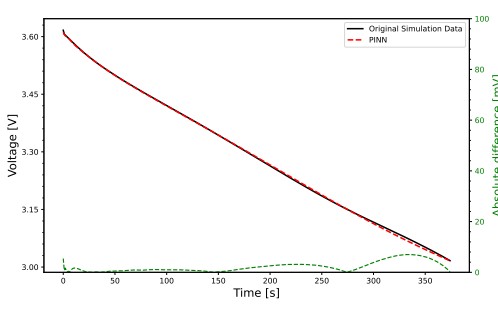
(a) The voltage changes over time.

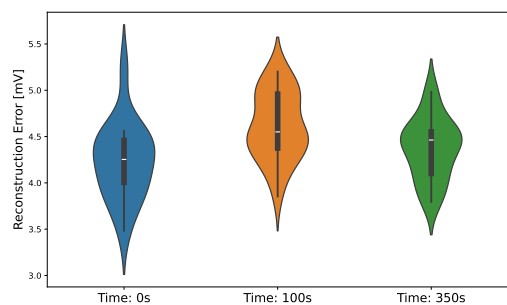
(b) Reconstruction voltage error on 0, 100 and 350s.

Figure 3: Comparison of the reconstructed voltage response using the PINN model with simulation data, including the voltage difference (a). Additionally, the voltage reconstruction at times 0, 100 and 300 seconds is presented, along with a boxplot illustrating the voltage error over 50 trials (b).

which should be the best estimation that does not incorporate PDE functions. From Table 4, it is evident that there are significant differences in the mean absolute error (MAE) and mean squared error (MSE) across the various methods. The autoencoder yielded a MAE of $2.64 \times 10^{-3}$ and an MSE of $1.11 \times 10^{-5}$. In contrast, the models that included PDE functions (such as PINN-T, PINN-S, PINN-B, and PINN-L for different layer width from tiny to large) demonstrated superior performance in both MAE and MSE. Notably, the PINN-L method achieved a MAE of $2.14 \times 10^{-3}$ and an MSE of $8.50 \times 10^{-6}$, showcasing its advantage in voltage reconstruction. Figure 4 illustrates the voltage changes over time, comparing the original simulation data, the autoencoder, and the various PINN methods. It can be observed that the PINN methods effectively capture the trend of voltage changes, with their reconstruction curves showing a significantly closer fit to the original data than the autoencoder. This indicates that the incorporation of PDE functions in the network structure enables a more effective capture of the dynamic characteristics of voltage changes, thereby improving reconstruction accuracy.

| Method | MAE | MSE |
|---|---|---|
| Autoencoder | $2.64 \times 10^{-3}$ | $1.11 \times 10^{-5}$ |
| PINN-T | $3.56 \times 10^{-3}$ | $2.13 \times 10^{-5}$ |
| PINN-S | $3.23 \times 10^{-3}$ | $1.56 \times 10^{-5}$ |
| PINN-B | $3.06 \times 10^{-3}$ | $3.38 \times 10^{-5}$ |
| PINN-L | $2.14 \times 10^{-3}$ | $8.50 \times 10^{-6}$ |

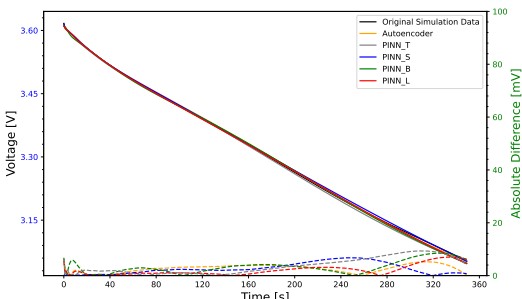

Figure 4: Comparison of voltage changes over time between the original simulation data and the reconstructed results from various methods, including the autoencoder and different PINN configurations (PINN-T, PINN-S, PINN-B, and PINN-L). The PINN methods demonstrate a closer fit to the original data, highlighting their effectiveness in capturing the dynamic characteristics of voltage changes.

## 7    CONCLUSION

In conclusion, our paper presents a novel method supported by PyBaMM for data generation and Physics-Informed Neural Networks (PINN) in modeling with the original P2D model. Compared to the Equivalent Circuit Model (ECM) and the Single Particle Model (SPM), the Pseudo-2D (P2D) model offers a more precise representation of the microscopic structure of batteries. Our results demonstrate superior performance in both voltage reconstruction and parameter estimation compared to alternative methods. We also envision that more advanced and complex neural network architectures could further enhance the resolution of challenges involving PDEs under the P2D framework.

**Ethics Statement:** This work focuses on parameter estimation for the pseudo-two-dimensional (P2D) lithium-ion battery model and voltage reconstruction. Upholding scientific integrity and transparency, we fully detail all experiments and open-access datasets in this paper. Practitioners using this work must audit their data and deployment contexts to ensure the application is fair and avoids negative societal consequences.

**Reproducibility Statement:** We include source code for a toy example in the supplementary materials to showcase the core functionality of our proposed method. This paper furnishes all essential details, such as specific hyperparameters, descriptions of the datasets, and our data processing pipeline. Upon the paper's acceptance, we commit to releasing the complete, well-documented codebase on a public repository. This ensures thorough verification of our work and aims to facilitate future research in the field.

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

## A  APPENDIX

**LLM Statement**  In this work, we leveraged Large Language Models (LLMs) as a supportive tool to assist with the writing and refinement of the manuscript. Specifically, their application was focused on enhancing the clarity, coherence, and precision of textual content—for instance, refining sentence structure to improve readability, ensuring consistent scientific phrasing across sections, and polishing descriptive passages to align with academic writing conventions. It is important to note that the use of LLMs was strictly limited to editorial and stylistic support; all core scientific content, including experimental designs, model formulations, data analyses, and key conclusions, was independently conceptualized, validated, and finalized by the authors to uphold the integrity and originality of the research.

### A.1  PYTHON BATTERY MATHEMATICAL MODELING

The Python Battery Mathematical Modeling (PyBaMM) framework (Sulzer et al., 2021) serves as a powerful tool for fast and flexible simulations of various battery models. By establishing a modular architecture, PyBaMM enhances collaboration and increases research impact within the field of battery modeling. This framework allows users to seamlessly integrate existing tools or develop new ones to address continuum models for batteries.

To achieve this, PyBaMM distinctly separates the components of models, discretization, and solvers, providing unparalleled flexibility for end users. This design enables the incorporation of new models, alternative spatial discretizations, or innovative time-stepping algorithms through a unified interface. Consequently, any additions can be immediately utilized with the existing suite of implemented models, facilitating comparisons across different models, discretizations, or algorithms while maintaining consistent variables such as hardware, software, and implementation details.

Moreover, PyBaMM's extensible "plug-and-play" submodel structure allows for the integration of additional physical phenomena into existing models. This flexibility eliminates the necessity to start from scratch when investigating new effects, enabling researchers to simultaneously explore a range of extensions to standard battery models. For example, it becomes straightforward to couple multiple degradation mechanisms to study their combined effects. The framework is supported by a comprehensive suite of tests that ensure robustness, allowing for the continuous addition of new models and solvers within an open-source environment.

PyBaMM is a key component of the Faraday Institution's "Common Modelling Framework," which is part of the Multi-Scale Modelling Fast Start project. This initiative aims to serve as a central repository for battery modeling research in the UK. To date, PyBaMM has been instrumental in the development and comparison of reduced-order models for lithium-ion (Marquis et al., 2019) and lead-acid (Sulzer et al., 2021; 2019) batteries. It has also been used to parameterize lithium-ion cells (Chen et al., 2020), model spirally-wound batteries (Tranter et al., 2020), analyze two-dimensional distributions in current collectors (Marquis et al., 2020; Timms et al., 2021), and investigate solid-electrolyte interphase (SEI) growth (Salinas & Kowal, 2020).

As collaborations with other members of the modeling community continue, both within and beyond the Faraday Institution, further research outcomes are anticipated. For an up-to-date list of publications that utilize PyBaMM, please visit the following link: `https://PyBaMM.org/publications/`.

For the details of PyBaMM model, we give the setting of real simulation parameter, such as the diffusion coefficient and conductivity for solid and electrode in cathode and anode. We use PyBaMM to generate more data for training and select 80 percent of these as training set. In the data generated by PyBaMM, we can directly produce data that includes a time dimension, along with all spatial information from the P2D model. Therefore, the generated data can be readily used for training our models. Some model variables available for PyBaMM models are shown in Table 3 (Part of All Variables). What's more, for the evaluation of PyBaMM model, we give some validations of voltage change information which are shown in Fig. 5.

A.2  BOUNDARY CONDITION OF PDE EQUATIONS

**Electrode mass transport**  The boundary conditions are:

$$D_{\mathrm{s}}^{\pm}\frac{\partial c_{\mathrm{s}}^{\pm}}{\partial r}(x,0,t) = 0,$$

$$D_{\mathrm{s}}^{\pm}\frac{\partial c_{\mathrm{s}}^{\pm}}{\partial r}\left(x,R_{\mathrm{s}}^{\pm},t\right) = -j_n^{\pm}(x,t),$$

$$c_{\mathrm{s}}^{\pm}(x,r,0) = c_{s,0}^{\pm},$$

where $c_{\mathrm{s}}(x,r,0)$ is the concentration of lithium ions in electrodes and $t=0, 0<r<R_{\mathrm{s}}^{\pm}$.

Table 3: Summary of parameters in P2D model.

| Category | Parameter | Unit |
|---|---|---|
| Basic Parameters | Time | $s$ |
| | $X_n$ | $m$ |
| | $X_s$ | $m$ |
| | $X_p$ | $m$ |
| | $r_n$ | $m$ |
| | $r_p$ | $m$ |
| Current/Voltage | Current | $A$ |
| | Voltage | $V$ |
| | Negative electrolyte potential | $V$ |
| | Positive electrolyte potential | $V$ |
| | Negative electrode potential | $V$ |
| | Positive electrode potential | $V$ |
| Battery Parameter | Negative electrode active material volume fraction | 1 |
| | Positive electrode active material volume fraction | 1 |
| | Negative electrode surface area to volume ratio | $m^{-1}$ |
| | Positive electrode surface area to volume ratio | $m^{-1}$ |
| Electrolyte parameters | Electrolyte concentration | $mol \cdot m^{-3}$ |
| | X-averaged electrolyte concentration | $mol \cdot m^{-3}$ |
| | Total lithium in electrolyte | $mol$ |

**Electrolyte mass transport**    For the boundary conditions are:

$$\frac{\partial c_{\mathrm{e}}^-}{\partial x}\left(0^-, t\right) = \frac{\partial c_{\mathrm{e}}^+}{\partial x}\left(0^+, t\right) = 0,$$

$$D_{\mathrm{e}}^{-,\mathrm{eff}}\left[c_{\mathrm{e}}\left(L^-\right)\right]\frac{\partial c_{\mathrm{e}}^-}{\partial x}\left(L^-, t\right) = D_{\mathrm{e}}^{\mathrm{sep,eff}}\left[c_{\mathrm{e}}\left(0^{\mathrm{sep}}\right)\right]\frac{\partial c_{\mathrm{e}}^{\mathrm{sep}}}{\partial x}\left(0^{\mathrm{sep}}, t\right),$$

$$D_{\mathrm{e}}^{\mathrm{sep,eff}}\left[c_{\mathrm{e}}\left(L^{\mathrm{sep}}\right)\right]\frac{\partial c_{\mathrm{e}}^{\mathrm{sep}}}{\partial x}\left(L^{\mathrm{sep}}, t\right) = D_{\mathrm{e}}^{+,eff}\left[c_{\mathrm{e}}\left(L^+\right)\right]\frac{\partial c_{\mathrm{e}}^+}{\partial x}\left(L^+, t\right),$$

$$c_{\mathrm{e}}\left(L^-, t\right) = c_{\mathrm{e}}\left(0^{\mathrm{sep}}, t\right),$$

$$c_{\mathrm{e}}\left(L^{\mathrm{sep}}, t\right) = c_{\mathrm{e}}\left(L^+, t\right),$$

$$c_{\mathrm{e}}^{\pm}(x, 0) = c_{\mathrm{e},0}^{\pm}.$$

**Solid potential of particles**    where the boundary conditions are:

$$-\sigma^{\mathrm{eff}}\frac{\partial \phi_{\mathrm{s}}^-}{\partial x}\left(0^-, t\right) = -\sigma^{\mathrm{eff}}\frac{\partial \phi_{\mathrm{s}}^+}{\partial x}\left(0^+, t\right) = \frac{I(t)}{A},$$

$$\frac{\partial \phi_{\mathrm{s}}^-}{\partial x}\left(L^-, t\right) = \frac{\partial \phi_{\mathrm{s}}^+}{\partial x}\left(L^+, t\right) = 0.$$

**Electrolyte potential**    The boundary conditions:

$$\phi_{\mathrm{e}}^-\left(0^-, t\right) = \phi_{\mathrm{e}}^+\left(0^+, t\right) = 0,$$

$$\phi_{\mathrm{e}}^-\left(L^-, t\right) = \phi_{\mathrm{e}}^{\mathrm{sep}}\left(0^{\mathrm{sep}}, t\right),$$

$$\phi_{\mathrm{e}}^{\mathrm{sep}}\left(L^{\mathrm{sep}}, t\right) = \phi_{\mathrm{e}}^+\left(L^+, t\right).$$

## A.3    THE PROVE OF P2D MODEL (PDE EQUATIONS)

### A.3.1    ELECTRODE MASS TRANSPORT

Assume the motion of ions in the electrode is governed by diffusion, described by Fick's laws and mass conservation principles.

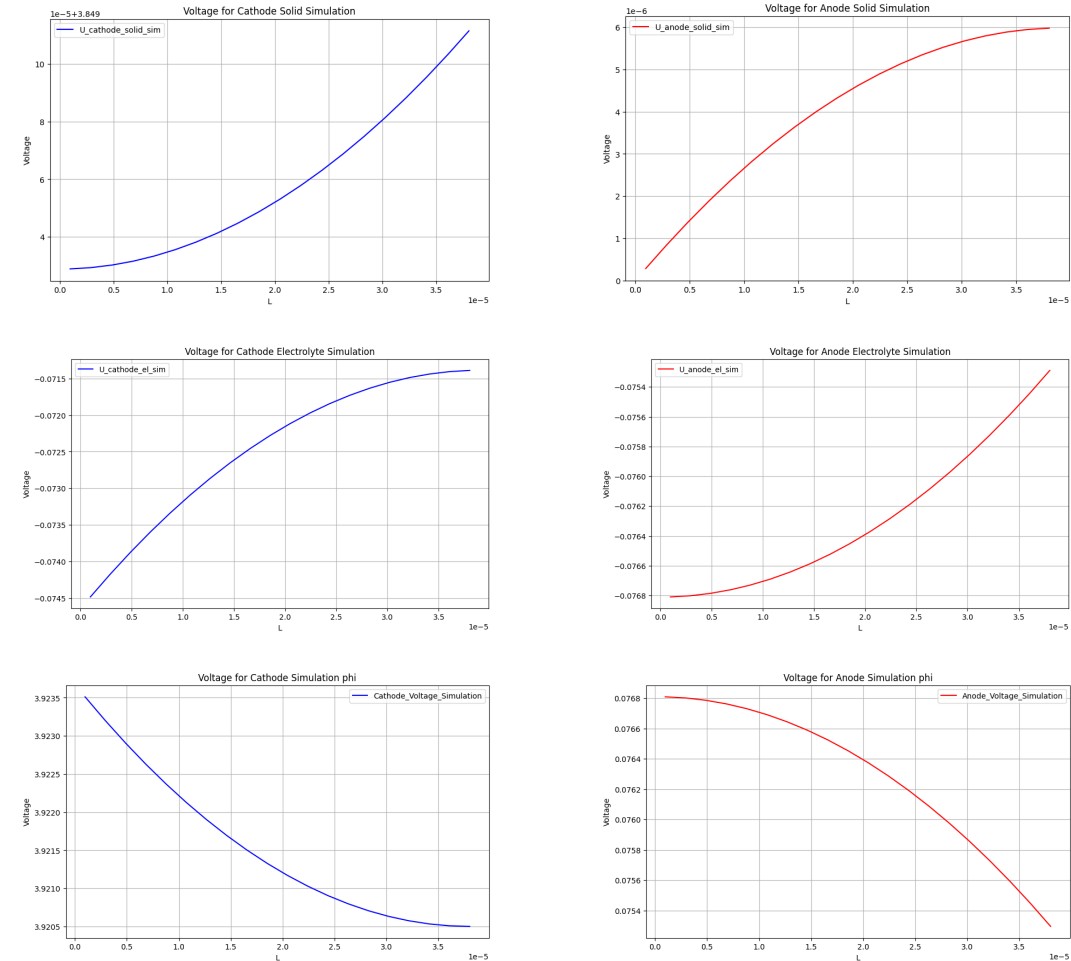

Figure 5: Validation of generating data given by PyBaMM. The left graph illustrates the voltage changes over time in the cathode, while the right graph displays the voltage changes over time in the anode.

The molar flux is proportional to the concentration gradient in the radial direction:

$$N_s^{\pm} = -D_s^{\pm} \frac{\partial c_s^{\pm}}{\partial r}$$

where $N_s^{\pm}(x, r, t)$ is molar flux of ions $(\text{mol} \cdot \text{m}^{-2} \cdot \text{s}^{-1})$, $D_s^{\pm}$ is diffusion coefficient $(\text{m}^2 \cdot \text{s}^{-1})$, $c_s^{\pm}(x, r, t)$ is ion concentration $(\text{mol} \cdot \text{m}^{-3})$, $\frac{\partial c_s^{\pm}}{\partial r}$ is radial concentration gradient.

Consider a spherical volume element with radius $r$ and thickness $dr$. The net accumulation rate of ions in the volume equals the difference between inward and outward fluxes:

$$\frac{\partial c_s^{\pm}}{\partial t} \cdot 4\pi r^2 dr = \left[ N_s^{\pm}(r) \cdot 4\pi r^2 \right] - \left[ N_s^{\pm}(r + dr) \cdot 4\pi (r + dr)^2 \right]$$

Dividing through by $4\pi dr$ simplifies to:

$$\frac{\partial c_s^{\pm}}{\partial t} \cdot r^2 = \frac{N_s^{\pm}(r) \cdot r^2 - N_s^{\pm}(r + dr) \cdot (r + dr)^2}{dr}$$

The right-hand side represents the negative radial derivative of the flux-term product:

$$\frac{\partial c_s^{\pm}}{\partial t} \cdot r^2 = -\frac{\partial}{\partial r} \left[ N_s^{\pm} \cdot r^2 \right]$$

Substitute $N_s^{\pm} = -D_s^{\pm} \frac{\partial c_s^{\pm}}{\partial r}$ into the continuity equation:

$$\frac{\partial c_s^{\pm}}{\partial t} \cdot r^2 = -\frac{\partial}{\partial r}\left[-D_s^{\pm}\frac{\partial c_s^{\pm}}{\partial r} \cdot r^2\right]$$

Simplifying the right-hand side:

$$\frac{\partial c_s^{\pm}}{\partial t} \cdot r^2 = \frac{\partial}{\partial r}\left[D_s^{\pm} \cdot r^2 \cdot \frac{\partial c_s^{\pm}}{\partial r}\right]$$

Dividing both sides by $r^2$ gives the spherical coordinate diffusion equation:

$$\boxed{\frac{\partial c_s^{\pm}}{\partial t}(x,r,t) = \frac{1}{r^2}\frac{\partial}{\partial r}\left[D_s^{\pm}r^2\frac{\partial c_s^{\pm}}{\partial r}(x,r,t)\right]}$$

### A.3.2 ELECTROLYTE MASS TRANSPORT

The derivation is rooted in the **mass conservation law** for lithium ions in the electrolyte, which states the rate of lithium ion accumulation in the control volume equals the net diffusion inflow rate plus the reaction-generated rate. This relationship describes how the lithium ion concentration in the electrolyte evolves over time, considering both diffusion transport and electrochemical reactions.

A micro-control volume is selected within the $j$-th phase ($j \in \{\pm, \text{sep}\}$), with:

- Cross-sectional area perpendicular to $x$-direction: $A$;
- Thickness along $x$-direction: $dx$;
- Total volume: $dV = A \cdot dx$;
- Electrolyte volume (accounting for porosity $\varepsilon_e^j$): $\varepsilon_e^j \cdot dV = \varepsilon_e^j \cdot A \cdot dx$.

The total amount of $Li^+$ in the electrolyte within the control volume is:

$$n_{Li^+} = c_e^j \cdot \varepsilon_e^j \cdot A \cdot dx$$

Differentiating with respect to time $t$ gives the accumulation rate of $Li^+$:

$$\frac{\partial n_{Li^+}}{\partial t} = \varepsilon_e^j \cdot A \cdot dx \cdot \frac{\partial c_e^j}{\partial t}$$

Dividing by the control volume $dV = A \cdot dx$ (to get the rate per unit volume) yields the LHS of the target equation:

$$\text{LHS} = \varepsilon_e^j \cdot \frac{\partial c_e^j}{\partial t}(x,t)$$

$Li^+$ diffusion in the electrolyte follows Fick's First Law (Fick, 1855), where the diffusion flux $N_e$ (amount per unit area per unit time) is:

$$N_e = -D_e^{\text{eff}} \cdot \frac{\partial c_e^j}{\partial x}$$

The negative sign indicates diffusion from high to low concentration:

- Inflow of $Li^+$ at $x$: $N_e(x) \cdot A$;
- Outflow of $Li^+$ at $x + dx$: $N_e(x+dx) \cdot A$.

Using Taylor expansion ($N_e(x+dx) \approx N_e(x) + \frac{\partial N_e}{\partial x} \cdot dx$), the net inflow rate is:

$$\text{Net inflow rate} = A \cdot [N_e(x) - N_e(x+dx)] \approx -A \cdot \frac{\partial N_e}{\partial x} \cdot dx$$

Substituting $N_e$ and dividing by $dV$ gives RHS1:

$$\text{RHS1} = \frac{\partial}{\partial x}\left(D_e^{\text{eff}} \cdot \frac{\partial c_e^j}{\partial x}(x,t)\right)$$

At the electrode-electrolyte interface, electrochemical reactions generate $Li^+$ in the electrolyte. Key relationships:

- Molar rate of Li$^+$ per unit interface area (Faraday's Law): $\frac{j_n^j}{F}$;
- Fraction of Li$^+$ remaining in electrolyte (correcting for migration): $(1 - t_c^0)$;
- Total rate per unit electrode volume (multiplying by specific surface area $a^j$):

$$\text{Reaction rate per unit volume} = a^j \cdot (1 - t_c^0) \cdot \frac{j_n^j}{F} \cdot F = a^j \cdot (1 - t_c^0) \cdot j_n^j(x, t).$$

Substituting the 4 equations, the electrolyte Li$^+$ concentration evolution equation is:

$$\boxed{\varepsilon_e^j \frac{\partial c_e^j}{\partial t}(x, t) = \frac{\partial}{\partial x}\left[D_e^{\text{eff}}\frac{\partial c_e^j}{\partial x}(x, t)\right] + a^j\left(1 - t_c^0\right)j_n^j(x, t).}$$

### A.3.3 SOLID POTENTIAL OF PARTICLES

The derivation equation 3 is based on the **charge conservation law**, which states that the net current flowing into a control volume equals the rate of charge accumulation within that volume:

$$\text{Net current inflow} = \text{Rate of charge accumulation}$$

Mathematically, for a control volume with closed surface $S$ and volume $V$, this is expressed as:

$$\oiint_S \mathbf{i} \cdot \mathbf{n} \, dS = -\frac{d}{dt} \iiint_V \rho_v \, dV$$

where:

- $\mathbf{i}$ is the current density vector (A/m$^2$);
- $\mathbf{n}$ is the outward unit normal vector to surface $S$;
- $\rho_v$ is the volume charge density (C/m$^3$);
- The negative sign indicates that net inflow reduces the charge density inside the volume.

For the solid electrode phase, we make the following simplifying assumptions:

- All quantities vary only along the $x$-direction (electrode thickness), so we use $\frac{\partial}{\partial x}$ instead of the full gradient operator $\nabla$.
- The current density in the solid phase follows Ohm's law:

$$i_x = -\sigma^{\text{eff}}\frac{\partial \phi_s^{\pm}}{\partial x},$$

where $i_x$ is the current density in the $x$-direction, and the negative sign indicates current flows from high potential to low potential.

- In solid electrodes, charge redistribution is very fast, leading to quasi-equilibrium conditions where the volume charge density remains approximately constant:

$$\frac{d}{dt}\iiint_V \rho_v \, dV \approx 0.$$

Using the divergence theorem, we convert the surface integral of current density to a volume integral:

$$\oiint_S \mathbf{i} \cdot \mathbf{n} \, dS = \iiint_V \nabla \cdot \mathbf{i} \, dV.$$

For our 1D case, the divergence of the current density vector simplifies to:

$$\nabla \cdot \mathbf{i} = \frac{\partial i_x}{\partial x}.$$

Substituting the expression for $i_x$ from Ohm's law:

$$\nabla \cdot \mathbf{i} = \frac{\partial}{\partial x}\left(-\sigma^{\text{eff}}\frac{\partial \phi_s^{\pm}}{\partial x}\right).$$

Electrochemical reactions at the solid-electrolyte interface act as sources or sinks of charge in the solid phase. The total reaction current per unit volume of electrode is given by:

$$\text{Reaction current density per unit volume} = a^\pm j_n^\pm.$$

Converting this to charge per unit volume using Faraday's constant (relating charge to moles of electrons):

$$\text{Charge source term} = a^\pm F j_n^\pm.$$

Combining these results with charge conservation (and noting the negligible accumulation term), we get:

$$\frac{\partial}{\partial x}\left(\sigma^{\text{eff}}\frac{\partial \phi_s^\pm}{\partial x}\right) = a^\pm F j_n^\pm.$$

Rearranging terms gives us the target equation:

$$\boxed{\frac{\partial}{\partial x}\left[\sigma^{\text{eff}}\frac{\partial \phi_s^\pm}{\partial x}(x,t)\right] - a^\pm F j_n^\pm(x,t) = 0.}$$

### A.3.4 ELECTROLYTE POTENTIAL

This derivation focuses on the liquid electrolyte phase in porous electrodes of lithium-ion batteries, adopting a one-dimensional ($x$-direction) spatial simplification (consistent with the Newman P2D model: "radial porous + axial 1D"). Key components:

- Lithium ions ($Li^+$, cation, $z_+ = 1$);
- Electrolyte anions (e.g., $PF_6^-$, $z_- = -1$);
- Solvent (carbonate-based, no charge transfer/reaction, $s_0 = 0$);
- Electrode-electrolyte interface (single-electron reaction, $n = 1$).

The Core Assumptions are shown as:

- Isothermal ($T = $ constant) and isobaric ($p = $ constant), satisfying Gibbs-Duhem relation;
- No side reactions (only single-electron reactions, e.g., $LiC_6 \leftrightarrow C_6 + Li^+ + e^-$ at anode);
- Mean activity coefficient: $\frac{\partial \ln f_\pm}{\partial \ln c_e} = 0$ (concentration-insensitive);
- Effective conductivity: $\sigma_e^{\text{eff}} = \kappa \cdot \frac{\varepsilon}{\tau}$ ($\kappa$: bulk conductivity, $\varepsilon$: porosity, $\tau$: tortuosity);
- No convective current (total liquid current from ion migration only).

For any ion $i$, the electrochemical potential $\bar{\mu}_i$ (combines chemical potential and electric potential) is:

$$\bar{\mu}_i = \mu_i + z_i F \phi_e \tag{9}$$

where: - $\mu_i$: Chemical potential of ion $i$ (depends on $c_e, T$); - $z_i$: Ion valence ($z_+ = 1$ for $Li^+$, $z_- = -1$ for anions); - $F$: Faraday constant (96485 C/mol); - $\phi_e$: Liquid electrolyte potential (core variable to solve).

For $Li^+$, chemical potential (ideal solution with activity correction) is:

$$\mu_{Li^+} = \mu_{Li^+}^\theta + RT\ln(c_e f_\pm) \tag{10}$$

where: - $\mu_{Li^+}^\theta$: Standard chemical potential (constant for fixed $T$); - $c_e$: $Li^+$ concentration in electrolyte (state variable); - $R$: Gas constant (8.314 J/(mol·K)); - $f_\pm$: Mean activity coefficient (constant, from Assumption 3).

Take $x$-direction gradient (1D transport) and substitute $\frac{\partial \ln f_\pm}{\partial x} = 0$:

$$\nabla \mu_{Li^+} = RT\nabla \ln c_e \tag{11}$$

For $Li^+$, migration current density $i_{Li^+}$ is:

$$i_{Li^+} = z_+ F D_+ c_e \left( -\nabla \ln c_e - \frac{z_+ F}{RT} \nabla \phi_e \right) \tag{12}$$

where $D_+$: $Li^+$ diffusion coefficient.

For anions, migration current density $i_-$ is:

$$i_- = z_- F D_- c_- \left( -\nabla \ln c_- - \frac{z_- F}{RT} \nabla \phi_e \right) \tag{13}$$

By electroneutrality ($c_e = c_-$, $z_+ c_e + z_- c_- = 0$) and $D_- \ll D_+$, simplify to $c_- \approx c_e$, $z_- = -1$.

Cation transference number $t_c^0$ (ratio of $Li^+$ current to total liquid current $i_e$):

$$t_c^0 = \frac{i_{Li^+}}{i_e} \quad (i_e = i_{Li^+} + i_-) \tag{14}$$

Substitute Eqs. (4)-(5) and electroneutrality into Eq. (6), then use $\sigma_e^{\text{eff}} = F^2(z_+^2 D_+ c_e + z_-^2 D_- c_-) \cdot \frac{\varepsilon}{\tau}$ (effective conductivity formula) to get:

$$i_e = -\sigma_e^{\text{eff}} \nabla \phi_e - \frac{\sigma_e^{\text{eff}} RT (1 - t_c^0)}{F} \nabla \ln c_e \tag{15}$$

Current continuity equation: *Current divergence = charge generation/consumption rate*

$$\nabla \cdot i_e = -a^\pm F j_n^\pm \tag{16}$$

where: - LHS: $\nabla \cdot i_e = \frac{\partial i_e}{\partial x}$ (1D simplification, current variation rate); - RHS: $a^\pm$: Electrode-electrolyte specific surface area; $j_n^\pm$: Interfacial reaction current density (negative sign = current consumption).

Substitute Eq. (7) into Eq. (8), apply product rule $\frac{\partial}{\partial x}(A \cdot B) = \frac{\partial A}{\partial x} B + A \frac{\partial B}{\partial x}$ to LHS:

$$\frac{\partial}{\partial x} \left( -\sigma_e^{\text{eff}} \nabla \phi_e - \frac{\sigma_e^{\text{eff}} RT (1 - t_c^0)}{F} \nabla \ln c_e \right) = -a^\pm F j_n^\pm \tag{17}$$

Move negative sign from LHS to RHS, rearrange terms:

$$\frac{\partial}{\partial x} \left( \sigma_e^{\text{eff}} \frac{\partial \phi_e}{\partial x} \right) + \frac{RT (1 - t_c^0)}{F} \frac{\partial}{\partial x} \left( \sigma_e^{\text{eff}} \frac{\partial \ln c_e}{\partial x} \right) = a^\pm F j_n^\pm \tag{18}$$

Adjust term order to get the *target formula*:

$$\boxed{\frac{\partial}{\partial x} \left( \sigma_e^{\text{eff}} \frac{\partial \phi_e}{\partial x}(x,t) \right) = -a^\pm F j_n^\pm(x,t) + \frac{2RT \left( 1 - t_c^0 \right)}{F} \frac{\partial}{\partial x} \left[ \sigma_e^{\text{eff}} \frac{\partial \ln c_e}{\partial x}(x,t) \right].} \tag{19}$$

