# OpenReview forum: "Fusing Deep Neural Networks with Multi-PDE Solvers: An Enhanced Approach to Battery Modeling"
_ICLR.cc/2026/Conference — ICLR 2026 Conference Withdrawn Submission_

### Official Review · Reviewer_nkEn · 2025-10-26

**Soundness:** 3
**Presentation:** 3
**Contribution:** 2
**Rating:** 4
**Confidence:** 4

**Summary:**

This paper employs a Physics-Informed Neural Network (PINN) to model the charge–discharge process of lithium-ion batteries, aiming to improve the computational efficiency of the original Pseudo-2D (P2D) model. The proposed approach is validated using simulated data generated from PyBaMM. However, the work lacks sufficient experimental validation and practical application, which limits its overall contribution.

**Strengths:**

The proposed approach is validated using simulated data generated from PyBaMM.

**Weaknesses:**

1.	The proposed method lacks real experimental validation. The proposed method has not been validated on real experimental data. The simulated data from PyBaMM are based on a simplified P2D model in which key parameters (e.g., x and r) are predefined. In practical applications, these parameters must be identified from real battery systems. The authors are encouraged to include experimental validation using real battery data to demonstrate the model’s practical effectiveness.

2.	The presented simulation results are too limited to fully demonstrate the model’s performance. In particular, Figures 3 and 4 show nearly linear voltage fitting results, which fail to highlight the capability of the proposed method. The authors should consider providing more complex or realistic simulation scenarios to better illustrate model accuracy and robustness.

3.	The method focuses primarily on parameter learning, without estimating critical state variables such as State of Charge (SOC) or State of Health (SOH). This omission limits its applicability in real-world battery management systems. The authors should explore the extension of their approach to downstream tasks relevant to battery health monitoring or control.

4.	The integration of a simplified P2D model with neural networks for model estimation is relatively straightforward and has been previously explored in the field. The paper would benefit from emphasizing any unique methodological or theoretical contributions beyond existing PINN-based approaches.

5.	Several typos should be corrected (e.g., line 174 on page 4 and line 361 on page 7).

**Questions:**

see above

---

### Official Review · Reviewer_J9Ho · 2025-10-27

**Soundness:** 2
**Presentation:** 2
**Contribution:** 2
**Rating:** 2
**Confidence:** 4

**Summary:**

This work proposes a machine learning framework for battery dynamics modeling that leverages a PINN formulation incorporating PDE constraints from the P2D model. The method achieves < 10 mV voltage reconstruction error on simulated P2D data and demonstrates improved parameter estimation compared to simple baseline models.

**Strengths:**

- The paper presents the full P2D equations, boundary conditions, and loss formulation, including detailed derivations that enhance completeness and transparency.
- The choice to decouple electrodes during PINN training is a sound design decision that simplifies the learning objective and improves model stability.

**Weaknesses:**

- The proposed method is a straightforward application of the PINN framework to the established P2D battery model. While the work may interest the battery modeling community, it offers limited novelty for the broader ML audience, as it primarily demonstrates an engineering application rather than methodological innovation.
- Furthermore, combining PINNs with P2D electrochemical models is not new. Prior works have addressed essentially the same problem, yet they are neither cited nor compared in this submission. For example:
	- PINN surrogate of Li-ion battery models for parameter inference. Part II: Regularization and application of the pseudo-2D model https://arxiv.org/abs/2312.17336
	- A physics-informed neural network approach to parameter estimation of lithium-ion battery electrochemical model https://www.sciencedirect.com/science/article/abs/pii/S0378775324012230
- Finally, the experimental evaluation is limited to a single 1C discharge protocol. No tests across different temperatures, C-rates, or cell chemistries are provided, despite claims of adaptability to various configurations. Broader validation is needed to substantiate these claims.

**Questions:**

- The encoder mapping from initial inputs (I, V, T) to electrochemical parameters may be ill-posed, as multiple parameter combinations can yield similar voltage profiles. Incorporating parameter constraints or auxiliary experimental measurements (e.g., conductivities from EIS) could help regularize estimation. What assumptions underlie the parameter estimation in Table 2, and how might such additional constraints be integrated into the current framework?
- Please report the wall-clock inference time of the proposed model compared with a standard PyBaMM P2D solver and, if possible, other neural surrogate models.
- Could the design choice of electrode decoupling be justified quantitatively through ablation studies or comparisons against a coupled-training alternative?

Minor typos/comments
- p. 4, Section 3.2.2: The the -> the
- p. 4, Section 3.2.3: $\sigma^\text{eff} = \varepsilon_e^{1.5} \sigma^e$ -> $\sigma^\text{eff} = \varepsilon_s^{1.5} \sigma^s$
- p. 4, eq (5): overpotential equation (first one) has an extra $F$, third equation has $k^\pm$ instead of $\kappa^\pm$ in Table 1
- p. 8, Table 2: for $D_s^-$, the target and estimated values are almost ten times different, while reported error is 4.36%. Is something going wrong?

---

### Official Review · Reviewer_qCtw · 2025-10-30

**Soundness:** 3
**Presentation:** 3
**Contribution:** 2
**Rating:** 4
**Confidence:** 4

**Summary:**

This paper introduces a hybrid approach of combing PINN with PDE solvers to model a cyle of battery charge (or discharge).

**Strengths:**

- An interesting view to combine PINN with PDE solvers
- Successful demonstrations on simulated battery charging data

**Weaknesses:**

After reading the whole paper, I am still confused in the following aspects.

- Why we need a hybrid approach rather than a pure PINN?
- What are the unique benefits when substituing P2D model in PyBAMM with your approach?
- How does the modeling efficiency being improved? I only observed the experimental results of reconstruction quality but not your claimed improved efficiency.
- Is the efficiency of P2D model a critical problem in battery modeling? I opted for 'No' because given the PyBAMM implementation, a P2D model runs pretty fast. More important problems or efficiency issues may arise when introducing more complicated ODEs, such as using a full DFN model with many degradation factors considered, such as litihium plating (https://docs.pybamm.org/en/stable/source/examples/notebooks/models/lithium-plating.html), SEI growth (https://docs.pybamm.org/en/stable/source/examples/notebooks/models/SEI-on-cracks.html).
- How could your approach support those degradation factors, and how could it support long-term simulation across multiple cycles?

**Questions:**

See the weakness part.

---

### Official Review · Reviewer_CpaS · 2025-10-31

**Soundness:** 2
**Presentation:** 3
**Contribution:** 2
**Rating:** 4
**Confidence:** 3

**Summary:**

This paper proposes a physics-informed neural network (PINN) framework that fuses deep neural networks with the electrochemical pseudo–two-dimensional (P2D) model for lithium-ion batteries. The goal is to accelerate model computation while maintaining physical consistency and enabling data-driven parameter estimation. The authors embed the governing PDEs of the P2D model into the neural network loss function, leveraging automatic differentiation to enforce electrochemical constraints. The method aims to reconstruct voltage responses and estimate key physical parameters simultaneously.

**Strengths:**

- The work addresses a meaningful challenge in electrochemical modeling — integrating data-driven learning with physics-based P2D formulations. The fusion of neural networks with multi-PDE systems for battery modeling remains a relatively underexplored area.

-  The paper offers a systematic approach for embedding P2D equations into a neural framework, which could inspire subsequent research on hybrid physical–data-driven modeling in battery systems.

- Although the quantitative improvements are limited, the conceptual contribution has potential value for the development of real-time or embedded battery management models where full P2D solvers are too computationally expensive.

**Weaknesses:**

- Limited experimental complexity: The evaluation is conducted only under simple charge/discharge protocols, such as constant-current (CC) tests. The model’s predictive accuracy is already unsatisfactory in these cases, raising concerns about its robustness and generalization to more realistic dynamic protocols (e.g., dynamic load profiles, multi-step currents).

- Marginal performance improvement: Quantitative results show limited gains over existing data-driven or reduced-order models, suggesting that the current PINN setup may not effectively capture the coupled PDE dynamics.

- Insufficient discussion on computational efficiency: While acceleration is a key motivation, the paper provides limited quantitative analysis (e.g., runtime, parameter count, scalability).

- Lack of ablation or robustness analysis: The influence of loss weighting between physical and data terms, or the stability of training under varying boundary conditions, is not discussed.

**Questions:**

- Do you use DFN model in Pybamm?

- Have the authors tested the model under dynamic or complex current profiles? If not, what prevents generalization beyond simple CC protocols?

- Please provide quantitative evidence of computational acceleration (e.g., runtime per simulation, GPU hours, or speed-up factor over PyBaMM).

---

### Note · Authors · 2025-11-12

I have read and agree with the venue's withdrawal policy on behalf of myself and my co-authors.